# Effects of Different Heat Treatments on Yak Milk Proteins on Intestinal Microbiota and Metabolism

**DOI:** 10.3390/foods13020192

**Published:** 2024-01-06

**Authors:** Senbiao Shu, Rong Jing, Liang Li, Wenhan Wang, Jinchao Zhang, Zhang Luo, Yuanyuan Shan, Zhendong Liu

**Affiliations:** 1Food Science College, Tibet Agriculture & Animal Husbandry University, Nyingchi 860000, China; shu19980724@126.com (S.S.); 17339783501@163.com (R.J.); jwllok@sina.com (L.L.); 17604863694@163.com (J.Z.); luozhang1759@xza.edu.cn (Z.L.); 2Institute of Edible Fungi, Shanghai Academy of Agricultural Sciences, Shanghai 201403, China; wangwenhan@saas.sh.cn; 3School of Food Science and Engineering, Northwest A & F University, Xianyang 712199, China

**Keywords:** yak milk, protein oxidation, intestinal microbiota, metabolism

## Abstract

Dairy products are susceptible to modifications in protein oxidation during heat processing, which can lead to changes in protein function, subsequently affecting intestinal health. Despite being a unique nutritional source, yak milk has not been thoroughly examined for the effects of its oxidized proteins on intestinal microbiota and metabolism. Hence, this study employed different heat treatment methods (low-temperature pasteurization, high-temperature pasteurization, and high-temperature sterilization) to induce oxidation in yak milk proteins. The study then assessed the degree of oxidation in these proteins and utilized mice as research subjects. Using metagenomics and metabolomics methods, this study examined the structure of intestinal microbial communities and metabolic products in mice consuming oxidized yak milk. The results showed a decrease in carbonyl and total thiol contents of yak milk proteins after different heat treatments, indicating that heat treatment causes oxidation in yak milk proteins. Metagenomic analysis of mouse intestinal microbiota revealed significant changes in 66 genera. In the high-temperature sterilization group (H), key differential genera included Verrucomicrobiales, Verrucomicrobiae, Akkermansiaceae, and 28 others. The high-temperature pasteurization group (M) mainly consisted of *Latilactobacillus*, *Bacillus*, and *Romboutsia*. The low-temperature pasteurization group (L) primarily comprised of *Faecalibacterium*, *Chaetomium*, Paenibacillaceae, *Eggerthella*, Sordariales, and 33 others. Functionally, compared to the control group (C), the H group upregulated translation and energy metabolism functions, the L group the M group significantly upregulated metabolism of other amino acids, translation, and cell replication and repair functions. Based on metabolomic analysis, differential changes in mouse metabolites could affect multiple metabolic pathways in the body. The most significantly affected metabolic pathways were phenylalanine metabolism, vitamin B6 metabolism, steroid hormone biosynthesis, and pantothenate and CoA biosynthesis. The changes were similar to the functional pathway analysis of mouse metagenomics, affecting amino acid and energy metabolism in mice. In summary, moderate oxidation of yak milk proteins exhibits a positive effect on mouse intestinal microbiota and metabolism. In conclusion, yak milk has a positive effect on mouse intestinal microflora and metabolism, and this study provides a scientific basis for optimizing dairy processing technology and further developing and applying yak milk.

## 1. Introduction

As a crucial protein source for humans, dairy products undergo structural alterations in proteins during heat treatment in the processing stage, affecting their functionality and nutritional value [1]. Different heat treatment procedures can have an impact on its nutritional value, such as high temperature pasteurization may cause some degeneration of the protein, making some nutrients slightly reduced, but overall little effect; Ultra-high temperature transient sterilization can kill microbes more completely, a treatment that causes partial degeneration of proteins, but may make some nutrients more easily absorbed. Currently, dairy products are widely recognized for their quality, taste, nutritional value, and safety, being a vital part of the human diet [2]. The oxidative modification of proteins in dairy products plays a significant role in their function and stability [3]. Protein oxidative modification is a post-translational modification process, which mainly includes free radical oxidation, metal ion-catalyzed oxidation, and reactive oxygen-mediated oxidative modification. These modifications can change the physical and chemical properties of proteins, affect their spatial conformation and active states, thus regulating biological activities within the organism [4,5]. Heat treatment, a common oxidative modification and pretreatment method for dairy products [6], is a key step in their processing [7]. It can ensure the microbial safety of dairy products [8], while causing oxidative modification of proteins [9,10]. Research by Dash K K, et al. [11] indicates that heat treatment can kill and deactivate microbial colonies, thereby enhancing food safety. Liu H et al. [12] found that in casein, oxidized lysine is the most important type of oxidative modification based on redox proteomics study of heat-induced changes in milk protein oxidative modifications. Despite the importance of heat-induced modifications for certain dairy products, “over-processing” often occurs due to varying heat treatment temperatures, leading to the loss of dairy nutrients and sensory quality [13]. Nevertheless, moderate heat treatments such as pasteurization and short time ultra-high temperature treatment generally do not significantly alter most nutrients in dairy products [14].

The gut microbiota, a key component of the human microecosystem, is often referred to as the “second genome” or “second brain” [15]. It not only participates in bidirectional communication between the gut and the brain [16], but also plays a pivotal role in generating beneficial metabolites and regulating immune responses [17]. Research by Koh A, et al. [18] reveals that gut microbes can break down indigestible food components (e.g., dietary fiber), producing beneficial metabolites such as butyric acid, propionic acid, and acetic acid. These short-chain fatty acids have been found to maintain gut health. Belkaid Y, et al. [19] demonstrated that gut microbes could affect the differentiation of T cells, as well as promote the generation of regulatory T cells (Treg) and Th17 cells. Treg cells can inhibit excessive immune responses, protecting the body from self-attack, while Th17 cells play a defensive role in resisting pathogen invasion. Additionally, gut microbes contribute to maintaining intestinal barrier integrity, inhibiting intestinal pathogens, promoting vitamin synthesis, and eliminating exogenous toxins [20]. Therefore, preserving the diversity and balance of gut microbes is an important strategy for regulating host health.

Metagenomics is a method that investigates the entire genome of a microbial community, utilizing high-throughput sequencing technologies to unveil the intricate relationship between the gut microbiota and the host [21]. Metabolomics employs advanced technologies such as mass spectrometry and chromatography to precisely identify and quantify changes in metabolites, metabolic characteristics, and metabolic pathways within biological samples [22]. By combining these two -omics approaches, we can gain a more comprehensive understanding of the types, functions, and impacts of the gut microbiota. Chen Qu, et al. [23] explored the impact of fructose and sucrose on host health and metabolism using both metagenomics and metabolomics approaches. Their results showed that fructose could increase the abundance of Lactobacillus and enhance the metabolism of carbohydrates and the TCA cycle ability. In contrast to fructose, sucrose increased the abundance of Klebsiella and Escherichia coli, leading to more severe hepatic steatosis, enteropathy, obesity, and visceral fat accumulation.

Yak milk, a unique nutritious source of milk, has not been fully studied regarding the impact of its oxidized proteins on gut microbiota and metabolism. To explore the impact of yak milk proteins after different heat processing treatments on gut microbiota and metabolism, we applied various heat processing methods (i.e., low-temperature pasteurization, high-temperature pasteurization, and high-temperature sterilization) to oxidize yak milk proteins. The study examined the degree of oxidation in yak milk proteins using mice as subjects and employed metagenomics and metabolomics to analyze gut microbiota and metabolic products in mice fed with oxidatively modified yak milk. These findings not only contribute to the understanding of the impact of different heat processing methods on yak milk for the Tibetan people but also provide a scientific basis for optimizing dairy product processing technology and the broader development and application of yak milk.

## 2. Materials and Methods

### 2.1. Experiment Materials

#### 2.1.1. Sample Collection

The yak milk used in this study was sourced from the plateau ranch in Damxung County, Lhasa, Tibet Autonomous Region. The experimental animals used were specific pathogen-free (SPF) male mice, strain C57BL, aged 8–10 weeks, which were supplied by China Shaanxi Xincheng Longyuan Biotechnology Co., Ltd. (Xi’an, China).

#### 2.1.2. Material of Study

Trichloroacetic acid was purchased from Beijing Solarbio Company China (Beijing, China). Sodium lauryl sulfate (SDS), guanidine hydrochloride were purchased from Hangzhou Gaojing Fine Chemical Company, Hangzhou, China. 2-nitrobenzoic acid (DTNB) was purchased from Shanghai Sangon Biotech Company, Shanghai, China. Urea, ethyl acetate, ethanol were purchased from Hangzhou Hanno Chemical Company, Hangzhou, China. 2,4-dinitrophenylhydrazine (DNPH), Mass Spectrometry (MS) grade water, acetonitrile (ACN), formic acid, and methanol were purchased from Thermo Scientific Company, New York, NY, USA. 2-chlorophenylalanine was purchased from Hangzhou Aladdin Company, Hangzhou, China. Hydrochloric acid and ammonium formate were purchased from Beijing Sigma Company, Beijing, China. Phosphate buffer solution and anhydrous ethanol were purchased from China Shanghai Titan Technology Co., Ltd. (Shanghai, China). BCA kit was purchased from Shanghai Selleck Chemicals Company, Shanghai, China.

#### 2.1.3. Experimental Instruments

The nucleic acid electrophoresis apparatus was obtained from Beijing 61 Instrument Factory, Beijing, China. The microplate reader is from Tecan Austria GmbH manufacturer and the equipment model is spark. The Covaris Ultrasonic Disrupter was sourced from the Covaris S2 System in Massachusetts, USA. For fluorometric quantification, the Qubit system from Life Technologies in CA, USA was utilized. The Agilent 2100 Bioanalyzer was employed for analysis, and it’s from Agilent Technologies Co., Ltd., in the Santa Clara, CA, USA. The PCR Instrument used was from Bio-Rad in the USA. Sequencing was conducted using an Illumina sequencer from San Diego, CA, USA. The Agilent Fragment Analyzer 5400 Automatic Capillary Electrophoresis System is from Agilent Technologies Co., Ltd., in the Santa Clara, CA, USA. Enzyme labeling instrument was acquired from Austria Tecan Austria GmbH. The freezing centrifuge is from Hunan Xiangyi Laboratory Instrument Development Co., Ltd., Zhuzhou, China. Other equipment includes a mixer and vortex mixer, tissue grinder from Ningbo Xinzhi Biotech Co., Ltd., Ningbo, China. Ultrasonic cleaner from Shumei Xingye Co., Ltd., Guangzhou, China. Filter membrane from Shanghai Jin Teng, China and a liquid chromatograph and mass spectrometer from Shanghai Thermo Fisher Company, Shanghai, China.

### 2.2. Experimental Method

#### 2.2.1. Sample Preparation

The collected yak milk samples were centrifuged at 2149× *g* for 15 min at 4 °C to separate the upper fat layer and retain the lower skim milk layer. This study used fat-removed yak milk. They were then divided into four groups: control group (Control), low temperature pasteurization group (65 °C, 30 min), high temperature pasteurization group (90 °C, 10 min), and high temperature sterilization group (120 °C, 10 min). Finally, the sterilized samples were freeze-dried at −50 °C.

#### 2.2.2. Animal Breeding and Sample Collection

The research involved a total of 40 healthy male mice. All mice were fed in standard laboratory mouse chow containing soybean meal, fish meal, grain, corn, etc. for one week, and were tested under the same environmental conditions. The mouse room was kept at a controlled temperature of 22–24 °C, with relative humidity maintained at 50–60%. The experimental animal room followed a 12-h light and 12-h darkness cycle. To ensure balanced nutrition, the mice were given sterile feed, and their water source was sterile as well. After 7 days of adaptive feeding, the mice were randomly divided into a normal diet group (C), a low-temperature pasteurized yak milk protein-fed mouse group (65 °C) (L), a high-temperature pasteurized yak milk protein-fed mouse group (90 °C) (M), and a high-temperature sterilized yak milk protein-fed mouse group (120 °C) (H). Each group consisted of 10 mice, and, excluding the control group, all other groups were administered milk through gavage on the third day. The experimental animals in each group were gavaged once at 8:00 every day for 12 consecutive weeks. At the end of the 12-week period, three mice were randomly chosen from each group. Fresh morning mouse feces were collected for total DNA extraction and intestinal flora detection. Lastly, cecal contents were collected and stored at −80 °C [24]. Table 1 provides a detailed plan for each group.

#### 2.2.3. Determination of Carbonyl Content

The method of Mestdagh, Kerkaert, Cucu, and De Meulenaer [26,27] was used with slight modifications. Briefly, 0.2 mL of diluted sample solution was transferred to a 2 mL centrifuge tube, and mixed with 0.8 mL of 2 mol/L HCl (containing 10 mmol/L 2,4-dinitrophenylhydrazine). Meanwhile, 2 mol/L HCl was used as a blank control. After incubation at room temperature for 1 h, 0.4 mL of 40% trichloroacetic acid was added. The mixture was allowed to stand for 30 min, and then centrifuged at 10,000 r/min for 20 min at 4 °C. The precipitate was washed three times with a cleaning solution (ethanol/ethyl acetate (1:1)), and then dissolved in 1.0 mL of 0.02 mol/L phosphate buffer solution (pH 6.5, containing 6 mol/L guanidine hydrochloride). The absorbance was measured at 370 nm, and the molar extinction coefficient of 22,000 was used to calculate the carbonyl content per milligram of protein, which was expressed as nmol/mg of protein, according to the Formula (1). Finally, data analysis and plotting were performed. The protein concentration was detected using a BCA kit.
(1)$$\mathrm{Carbonyl}\;\mathrm{Content}\;(\mathrm{mol/mg})=\frac{OD1-OD2}{22\times C\times P}\times 125\times 10^{5}$$

OD1 measures the absorbance of the tube, OD2 is the absorbance of the opposite tube, C is the colorimetric light diameter, and P is the protein concentration in the sample.

#### 2.2.4. Determination of Total Thiol Content

The free and total thiol groups were determined using the DTNB colorimetric method, according to the method of Cao Y [28] et al., with minor modifications. A 0.8 mL defatted sample was mixed with 0.2 mL of 8 mol/L urea and 0.03 g/mL SDS solution and reacted in the dark for 1 h. Then, 0.5 mL of 40% (*w*/*v*) trichloroacetic acid solution was added, and the mixture was incubated for another 1 h. After centrifugation (5000 r/min, 10 min), the precipitate was collected and washed with 40% trichloroacetic acid, and the washing process was repeated twice under the same conditions. The precipitate was dissolved in 1 mL of 0.03 g/mL SDS solution, and 0.2 mL of DTNB solution was added and reacted in the dark for 1 h. The absorbance was measured at a wavelength of 412 nm, and a blank control group without DTNB was employed. Finally, the amount of thiol substances in each milligram of protein was expressed, with the unit being nmol/mg. It was calculated according to Formula (2):
(2)$$\mathrm{SH}\;(\mathrm{mol/mg})=73.53\times A_{412}\times \mathrm{D/C}$$where A_412_ is the absorbance value of the sample after removing the reagent blank, D—dilution multiple, C—protein concentration mg/mL.

#### 2.2.5. DNA Extraction and Sequencing

Microbial DNA in mouse fecal samples was extracted using the CTAB method [29]. The extracted samples were sequenced using the 16S rRNA gene sequencing method [30]. The Illumina NovaSeq sequencing platform model is Novaseq 6000, used for sample detection, library construction, and sequencing [31].

#### 2.2.6. Untargeted Metabolomics Metabolite Extraction

A sample weighing between 62.9 mg and 100 mg was taken using a one ten-thousandth balance and then homogenized with 0.6 mL of 2-chlorophenylalanine and methanol. Next, 100 mg of glass beads were added, and the sample was ground at 55 Hz for 90 s in a tissue grinder. The sample was then sonicated at room temperature for 10 min. After centrifugation at 12,000 rpm for 10 min at 4 °C, the supernatant was collected. Subsequently, 200 μL of the supernatant underwent filtration using a 0.22-μm filter, and the resulting filtrate was collected for further analysis. Finally, 20 µL of each sample was taken and mixed into the QC sample (QC: quality control, which was used to correct the deviation in the analysis of the mixed samples and the error caused by the instrument analysis itself). LC-MS detection was carried out on the samples.

##### Chromatographic Conditions

The Thermo Ultimate 3000 equipped with the ACQUITY UPLC^®^ HSS T3 1.8 µm (2.1 × 150 mm) chromatographic column was used in this study. The auto-sampler temperature was set to 8 °C. The gradient elution was performed at a flow rate of 0.25 mL/min, column temperature of 40 °C, and an injection volume of 2 μL. The mobile phase was 0.1% formic acid in water (C) and 0.1% formic acid in acetonitrile (D) for positive ions; and 5 mM ammonium formate in water (A) and acetonitrile (B) for negative ions. The gradient elution program was as follows: 0–1 min, 2% B/D; 1–9 min, 2–50% B/D; 9–12 min, 50–98% B/D; 12–13.5 min, 98% B/D; 13.5–14 min, 98−2% B/D; 14–20 min, 2% D—positive mode (14–17 min, 2% B—negative mode).

##### Mass Spectrometric Conditions

The electrospray ion source (ESI) was used in both positive and negative ionization modes. The spray voltage was 3.50 kV for positive ions and 2.50 kV for negative ions. Other parameters included sheath gas 30 arb, auxiliary gas 10 arb, capillary temperature of 325 °C, and a resolution of 70,000. The scan range was 81~1000, and HCD was used for secondary fragmentation with a collision voltage of 30 eV. Dynamic exclusion was used to eliminate unnecessary MS/MS information.

#### 2.2.7. Data Processing and Analysis

In this experiment, independent samples Tukey test was performed using SPSS 25.0 statistical software. *p* < 0.05 indicated that the difference was statistically significant. Other data were processed using mean ± standard deviation, and OriginPro 2012 was used for plotting. KneadData software was used for quality control (based on Trimmomatic) and host removal (based on Bowtie2) of the raw data. Before and after KneadData, FastQC was used to check the rationality and effect of quality control [32,33]. Starting from the quality-controlled reads and the reads with host genes removed, HUMAnN3 software (based on DIAMOND) was used to align the reads of each sample to the database (UniRef90). According to the UniRef90 ID and the corresponding relationship of each database, the annotation information and relative abundance table of each functional database were obtained [34,35,36,37]. The original data obtained from liquid chromatography-mass spectrometry (LC-MS/MS) was processed using Proteowizard software (v3.0.8789).

## 3. Results

### 3.1. The Effect of Different Heat Processing Treatments on the Carbonyl Content of Yak Milk Protein

The formation of carbonyl groups is one of the most significant changes following protein oxidation. Thus, the carbonyl content has become a common indicator to measure the degree of protein oxidation [38]. As shown in Figure 1, compared with the control group, the overall carbonyl content of the three groups that underwent different heat processing treatments was significantly increased (*p* < 0.05). The carbonyl content increased first and then decreased with the rise in temperature. These results indicate that different heat processing treatments can lead to different degrees of oxidation in yak milk proteins.

### 3.2. Effect of Different Thermoprocessing Treatments on Total Sulfhydryl Content of Yak Milk Protein

The reduction of thiol groups is one of the main common features of protein changes in response to oxidative [39]. As shown in Figure 2, we found that the total thiol content in the samples exposed to different heat treatments were decreased (*p* < 0.05), but the total thiol content gradually increased with the rise in temperature. The decrease in the total thiol content indicates that the protein is undergoing oxidation, and this oxidation process intensifies with rising temperatures.

### 3.3. Effect of Yak Milk Proteins on Mouse Intestinal Microbiota

#### 3.3.1. Statistics of the Sequencing Data

The raw data (Raw Data) generated from Illumina sequencing contains a certain proportion of low-quality data. To ensure the accuracy and reliability of subsequent analyses, it is necessary to preprocess the original sequencing data. This involves tasks such as quality control (Trimmomatic [32] Parameters: ILLUMINACLIP:adapters_path:2:30:10 SLIDINGWINDOW:4:20 MINLEN:50), and removal of host sequences (Bowtie2 [33]), to obtain valid sequences (clean data) for downstream analysis. For fecal sequencing using the Illumina NovaSeq, three mice were randomly chosen from each sample group. Through quality control and the removal of host sequences, valid sequences were obtained. Specific data statistics can be found in Table 2.

#### 3.3.2. Analysis of the Species Composition

To assess the species composition and diversity information of the samples, Kraken2 [40] was used to annotate and classify the valid sequences of all samples. Based on the Bracken [41] database, a basic species composition analysis was performed on the obtained data. For each sample, the ratio of the number of sequences to the total number of sequences at the Kingdom, Phylum, Class, Order, Family, Genus, and Species levels was statistically analyzed. Figure 3 shows the relative degree of annotation at each classification level in each sample. Notably, the species detected in these samples were as follows: Archaea (616), Bacteria (83,786,651), Fungi (6186), Picornaviridae (1,922,113), Leviviridae (93), Positiviridae (82), Subviridae (2), Sangerviridae (789), Viruses (80,875); and their corresponding proportions were: Archaea (0.00%), Bacteria (97.66%), Fungi (0.01%), Picornaviridae (2.24%), Leviviridae (0.00%), Positiviridae (0.00%), Subviridae (0.00%), Sangerviridae (0.00%), Viruses (0.09%).

As shown in Figure 4A, at the phylum level, Firmicutes, Actinobacteria, Bacteroidetes, and Verrucomicrobia are the dominant phyla in the fecal samples of each group of mice. In the C group, Firmicutes, Actinobacteria, Bacteroidetes and Verrucomicrobia accounted for 34.961%, 30.490%, 22.917% and 7.526%, respectively. In the H group, they accounted for 31.765%, 18.688%, 31.083% and 8.775%, respectively. In the L group, they accounted for 39.071%, 27.582%, 27.623% and 0.417%, respectively. In the M group, they accounted for 54.882%, 24.456%, 15.559% and 2.291%, respectively. Compared with the C group, the H group exhibited a decrease in the relative abundance of Firmicutes and Actinobacteria, while an increase in the relative abundance of Bacteroidetes and Verrucomicrobia; the L group had a decrease in the relative abundance of Actinobacteria and Verrucomicrobia, while an increase in the relative abundance of Firmicutes and Bacteroidetes; the M group showed a decrease in the relative abundance of Actinobacteria, Bacteroidetes and Verrucomicrobia, while an increase in the relative abundance of Firmicutes. In addition, the relative abundances of Bacteroidetes and Verrucomicrobia in the H group were higher than those in the L and M groups, the relative abundance of Actinobacteria in the L group was higher than that in the H and M groups, and the relative abundance of Firmicutes in the M group was higher than that in the H and L groups. This suggests that the intake of yak milk proteins after different heat treatments can affect the composition of the gut microbiota in mice at the phylum level.

As shown in Figure 4B, at the genus level, the gut microbial communities of all groups of mice were mainly composed of *Lactobacillus*, *Bifidobacterium*, *Muribaculum*, *Adlercreutzia*, *Duncaniella*, *Bacteroides*, *Limosilactobacillus*, *Ligilactobacillus*, etc. In the C group, these genera accounted for 20.258%, 20.560%, 11.113%, 4.333%, 4.709%, 4.449%, 3.160% and 7.430%, respectively. In the H group, they accounted for 17.833%, 8.571%, 15.227%, 4.571%, 7.322%, 5.173%, 1.598% and 7.311%, respectively. In the L group, they accounted for 18.855%, 8.722%, 8.033%, 17.373%, 10.291%, 4.886%, 8.175% and 3.200%, respectively. In the M group, they accounted for 36.956%, 15.068%, 5.658%, 5.740%, 4.727%, 2.684%, 9.942% and 4.845%, respectively. Compared with the C group, the relative abundances of *Bifidobacterium* and *Ligilactobacillus* were decreased in all three groups, while those of *Adlercreutzia* and *Duncaniella* were increased. Compared with the three groups, the relative abundances of *Muribaculum*, *Bacteroides*, and *Ligilactobacillus* in the H group were higher than those in the L and M groups; the relative abundances of *Adlercreutzia* and *Duncaniella* in the L group were higher than those in the H and M groups; and the relative abundances of *Lactobacillus*, *Bifidobacterium*, and *Limosilactobacillus* in the M group were higher than those in the H and L groups. This suggests that the intake of yak milk proteins after different heat treatments significantly alters the gut microbiota in mice.

#### 3.3.3. Analysis of Common Species in Each Sample

In the samples, we search for unique or shared species between groups based on the presence or absence of species. For experimental schemes with fewer groups (less than or equal to 5), we draw Venn diagrams [42] to analyze unique or shared species between different sample groups, which intuitively display the similarities and overlaps in species composition between sample groupings (Figure 5). According to the OTUs clustering results, there were 505 OTUs shared by the four groups; the C group had 141 unique OTUs, the H group had 146 unique OTUs, the L group had 134 unique OTUs, and the M group had 192 unique OTUs. Compared with the C group, the number of unique bacterial species increased in the H group, decreased in the L group, and significantly increased in the M group. This indicates that the intake of yak milk proteins processed with different heat processing treatments can change the number of gut microbial communities in mice and change their original gut microbiota structure.

To further analyze the differences in the microbial community structure between groups and identify species with abundance differences between groups, we used the LEfSe analysis method to identify the specific microbial species distributed in each group. The analysis results, including the evolutionary clade map (Figure 6) and LDA bar chart (Figure 7), visually demonstrate the distinctive species in each group with significant differences in community structure. We found that 66 bacterial genera changed, with the main difference in the C group being Phoenicibacter and Longibaculum; the H group had 28 different genera such as Verrucomicrobiales, Verrucomicrobiae, Akkermansiaceae, etc.; the L group had 33 different genera such as Faecalibacterium, Chaetomium, Paenibacillaceae, Eggerthella, Sordariales, etc.; the M group primarily consisted of Latilactobacillus, Bacillus, and Romboutsia.

#### 3.3.4. Analysis of Functional Relative Abundance and Metabolic Pathway Profile

The KEGG database was used to cluster genes with similar functions [43]. Based on the annotation results, a relative abundance statistical graph was constructed for each sample at various functional levels. As shown in Figure 8A, these functions include: Metabolism of other amino acids, Carbohydrate metabolism, Amino acid metabolism, Translation, Metabolism of cofactors and vitamins, Replication and repair, Energy metabolism, Biosynthesis of other secondary metabolites, Lipid metabolism, Glycan biosynthesis and metabolism, Xenobiotics biodegradation and metabolism, Drug resistance: antimicrobial, Cell growth and death, Nucleotide metabolism, Metabolism of terpenoids and polyketides, Membrane transport, Endocrine system, and Cancer: overview. Compared to group C, group H upregulated Translation and Energy metabolism functions and downregulated Amino acid metabolism. Group L upregulated Metabolism of other amino acids, Translation, and Replication and repair, downregulated Amino acid metabolism, and Metabolism of cofactors and vitamins. Group M significantly enhanced Metabolism of other amino acids, Translation, Replication and repair, and significantly downregulated Carbohydrate metabolism, Amino acid metabolism, Biosynthesis of other secondary metabolites, and Glycan biosynthesis and metabolism. Taken together, the functional impact of heat-processed yak milk proteins on mice is mainly associated with amino acid metabolism, cell repair, and energy metabolism.

Further functional metabolic pathway analysis was conducted on the distribution of gut microbiota. As illustrated in Figure 8B, the 20 primary affected metabolic pathways with significant abundance are Ribosome information processing (map03010), Lysine metabolism (map00473), Aminoacyl-tRNA biosynthesis (map00970), Carbon fixation in photosynthetic organisms (map00710), DNA mismatch repair (map03430), Amino acid metabolism: biosynthesis of valine, leucine, and isoleucine (map00290), Vancomycin resistance (map01502), Cell cycle—Caulobacter (map04112), Peptidoglycan biosynthesis (map00550), Folate biosynthesis (map00670), Amino acid biosynthesis (map01230), Protein export (map03060), Streptomycin biosynthesis (map00521), Glycolysis/Gluconeogenesis (map00010), Lysine biosynthesis (map00300), Homologous recombination (map03440), Alanine, aspartate, and glutamate metabolism (map00250), Drug metabolism—other enzymes (map00983), Pantothenate and CoA biosynthesis (map00770). Compared to group C, group H upregulated map03010, map00710, map03430, map01230, map00521, map00010, map00983 and downregulated map01230. Group L upregulated map03010, map00473, map00970, map03430, map03060, map00250 and downregulated map01230, map00300, map00770. Group M significantly enhanced map03010, map00473, map00970, map03430, map00290, map04112, map03060, map03440, map00250 and significantly downregulated map00710, map00550, map00670, map00521, map00010. Analysis of the corresponding metabolic pathways revealed that the most notable effects were observed in amino acid metabolism, biosynthesis, metabolism, and carbohydrate metabolism.

### 3.4. Effect of Yak Milk Proteins after Different Heat Processing Treatments on Mouse Metabolism

#### 3.4.1. Sample QC and Standardization Calibration

When employing chromatographic columns, there is a risk of contamination, which can introduce errors into the instrument’s chromatographic analysis. To ensure the accuracy of measurement data, it is imperative to standardize and perform quality control on the sample data. As shown in Figure 9A, the QC samples in the PCA plot are clustered together, indicating an effective correction. In Figure 9B, the median and quartiles of metabolite concentrations are uneven before standardization, but are basically on the same level after standardization.

#### 3.4.2. Metabolite Content Statistics

After correction of the metabolites, statistical analysis revealed differences between groups. The results are shown in Figure 10. The top 20 metabolites in terms of abundance are Oleamide, Fenpropimorph, 7-Ketolithocholic acid, Cholic acid, Monoolein, PPNicotinic acid, Linoleoyl ethanolamide, L-Phenylalanine, (+/−)12(13)-DiHOME, Isoleucine, Hexadecanamide, D-Proline, 2-Hydroxycinnamic acid, (E)-Octadecadienoic acid, Xanthine, L-Tyrosine, Oleoyl ethanolamide, MAG (18:3), Hypoxanthine, and Stearamide.

#### 3.4.3. Analysis of Differences among Samples

To investigate the differences between the samples and the correlations between the groups, a principal component analysis was first conducted between the groups. Then, the metabolites in the detected samples were clustered. As shown in Figure 11A, all points are within the ellipse and are relatively concentrated, demonstrating a good model fit and significant differences between groups. Figure 11B shows that the metabolites in each group of samples are clustered in different locations, indicating significant differences. Nevertheless, principal component analysis is an unsupervised analysis method and may not provide a compelling explanation for the differences between samples. Therefore, additional analysis is required for a more thorough understanding.

#### 3.4.4. Partial Least Squares Discriminant Analysis

Although the figures and principal component analysis have initially suggested differences between the samples, to validate these differences, permutation tests were employed. A permutation test statistic, often indicative of the model’s predictive power, was formulated. Using the permutation method, the distribution of this test statistic was derived—a random distribution shaped by chance factors. Comparing the observed test statistic of the sample with this distribution could ascertain whether the discriminant effect of our PLS-DA model resulted solely from random factors. As depicted in Figure 12A, the point clouds of different groups of samples are distributed in distinct areas, indicating that the PLS-DA model has a good discriminant effect and the presence of significantly different metabolites between the groups. Figure 12B illustrates the permutation test of PLS-DA, with prediction accuracy as the chosen test statistic. The observed test statistic resides on the right side of the random distribution, implying its significance (the observed value significantly surpasses the random value). The *p*-value is less than 0.05, indicating the significance of our PLS-DA discriminant model. This implies that the discriminant effect is substantial and can effectively distinguish between different groups, suggesting the existence of significantly different metabolites among them.

#### 3.4.5. Univariate Analysis

After our initial analysis, we identified differences in metabolites between the groups. However, understanding the physiological implications of these changes requires further investigation. To address this, we calculated the fold change (FC) of metabolite alterations, providing a quantitative measure of the magnitude of these changes. This information, coupled with *p*-values, was then used to filter out metabolites with significant effects, offering a better understanding of the impact on the body. As shown in Figure 13, metabolites in the yellow region have a *p*-value of less than 0.05, and an absolute change greater than 2. These metabolites exhibit significant differences between groups and large variations. To visually present the differences in metabolites between groups, we statistically analyzed the selected metabolites. Box plots of the top 25 differential metabolites are shown in Figure 14. Notably, these metabolites include 2,3-Dinor-8-epi-prostaglandin F2α, 4-(methylthio)-6-phenyl-2-(3-pyridyl)pyrimidine-5-carbonitrile, *N*-Acetyl-DL-serine, Indole, 2-Arachidonoyl glycerol, 3-Hydroxypicolinic acid, Purine, (R)-3-Hydroxy myristic acid, Reserpine, 4-(2-chloro-6-fluorobenzyl)-3,5-dimethyl-4-prop-2-ynyl-4H-pyrazole, All trans-Retinal, gamma-Glu-Leu, Acetanilide, 4-(1H-pyrazol-1-yl)-*N*,*N*-bis(2-pyridinylmethyl)benzenesulfonamide, Oxaceprol, Isoferulic acid, L-Pyroglutamic acid, 13-HPODE, ethyl 1-(4-acetyl-2-aminophenyl)piperidine-4-carboxylate, Boc-beta-cyano-L-alanine, 9-Oxo-10(E),12(E)-octadecadienoic acid, 15-Acetyldeoxynivalenol, methyl 2,5-dimethyl-1H-pyrrole-3-carboxylate, 9-Oxo-ODE, and Malvidin.

#### 3.4.6. Metabolic Pathway Analysis

Through enrichment analysis, we identified metabolites exhibiting significant differences between groups (*t*-test, *p* < 0.05). This approach allowed us to uncover essential biological pathways influencing specific biological processes, shedding light on the fundamental molecular mechanisms at play. Notably, we found key functional biological pathways within metabolic pathways. As shown in Figure 15A, the metabolic pathways were primarily enriched in steroid hormone biosynthesis, vitamin B6 metabolism, phenylalanine metabolism, pantothenate and CoA biosynthesis, primary bile acid biosynthesis, phenylalanine, tyrosine and tryptophan, lysine degradation, purine metabolism, pyrimidine metabolism, biosynthesis of unsaturated fatty acids, arginine and proline metabolism, nicotinate and nicotinamide metabolism, histidine metabolism, beta-alanine metabolism, alanine, aspartate and glutamate metabolism, glutathione metabolism, aminoacyl-tRNA biosynthesis, porphyrin and chlorophyll metabolism, glycine, serine and threonine metabolism, steroid biosynthesis, arachidonic acid metabolism, and tyrosine metabolism.

Following this, we performed a topology analysis to determine the most significant biological functions in these enriched metabolic pathways, emphasizing those most influenced by the metabolites. As shown in Figure 15B, the metabolic pathways in the blue region, which were significant in the ORA enrichment analysis, included phenylalanine metabolism, vitamin B6 metabolism, steroid hormone biosynthesis, and pantothenate and CoA biosynthesis.

## 4. Discussion

In this study, we observed notable changes in the gut microbiota composition and metabolic products of mice following the consumption of yak milk proteins subjected to various heat treatments. These observed changes hold the potential to similarly impact human gut health.

Based on metagenomic analysis, at the phylum level, Firmicutes were found to be the most abundant, followed by Bacteroidetes and Verrucomicrobia—a distribution resembling that of the human gut microbiota [43]. At the genus level, the most significant changes in the mouse gut microbiota were found in Latilactobacillus, Bacteroides, and Akkermansia. Latilactobacillus exhibited potential benefits in enhancing gastrointestinal function, controlling endotoxins, inhibiting the growth of decay-causing bacteria, and boosting immunity [44]. Bacteroides, encoding a higher number of carbohydrate-degrading enzymes compared to Firmicutes [45], indicated a decline in immunity when their abundance decreased, affecting gut immunity. Akkermansia plays a crucial role in promoting gut immunity, mitigating gut inflammation, and demonstrating potent anticancer effects [46]. In response to a decline in immunity, Akkermansia abundance rapidly increased, supporting gut immunity, maintaining microbiota diversity, and promoting gut health. Moreover, results from OTUs clustering provided additional evidence supporting the impact of differently heated, oxidatively modified yak milk proteins on gut microbiota diversity in mice. Analysis of functional abundance and metabolic pathways revealed significant influences on amino acid metabolism, energy metabolism, carbohydrate metabolism, and transcription in mice. Therefore, yak milk proteins, undergoing functional changes due to distinct heat treatments, have considerably affected the structure and function of the mouse gut microbiota, along with their metabolism and health status.

Metabolomic studies revealed that the consumption of yak milk proteins, which have undergone various heat treatments, significantly regulated various metabolites in mice. These metabolites included phenylalanine, Vitamin B6, steroidal hormones, pantothenate, and Coenzyme A, each playing pivotal roles in the body. Phenylalanine, an essential amino acid in the human body, typically converts into tyrosine via phenylalanine hydroxylase. Tyrosine and phenylalanine jointly synthesize critical neurotransmitters and hormones, contributing to glucose and fat metabolism. Insufficient phenylalanine may impede tyrosine synthesis, leading to decreased thyroid hormone levels and affecting overall metabolic function. Vitamin B6, a constituent of certain coenzymes, participates in diverse metabolic reactions, particularly those involving amino acid metabolism. The metagenomic studies revealed disruptions in amino acid metabolism in mice, possibly linked to altered Vitamin B6 metabolism. Steroidal hormones, also known as steroid hormones, play vital roles in maintaining life, regulating reproductive functions, body development, immune regulation, and treating skin diseases [47]. Pantothenic acid, also known as Vitamin B5, is formed through the enzymatic reaction of α-ketoisovaleric acid and L-aspartic acid [48]. Coenzyme A, comprising of pantothenic acid, adenine, ribonucleic acid, and phosphate, serves as an acyl carrier in enzymatic reactions [49]. In cells, Coenzyme A combines with acetic acid to form Acetyl-CoA, which enters the oxidation process. Acetyl-CoA plays a critical role in many metabolic pathways, such as glycolysis, the β-oxidation of fatty acids, and the citric acid cycle [50]. Therefore, the most significant metabolic pathways in mice were amino acid metabolism and energy metabolism. In summary, different heat treatments of oxidatively modified yak milk proteins can affect the structure and metabolism of the mouse gut microbiota, thereby influencing various functions such as amino acid metabolism, cell repair, and energy metabolism.

In the dietary habits of the Tibetan people, yak milk products hold a crucial role. Through the application of heat treatments, oxidatively modified yak milk proteins are introduced into dairy products, potentially influencing the composition and function of their gut microbiota, including important bacteria like Latilactobacillus and Akkermansia. These bacteria play key roles in maintaining gut health and boosting immunity. Metabolomic studies have highlighted the significance of phenylalanine in neurotransmitter and hormone synthesis. Different heat treatments may impact the phenylalanine content in yak milk proteins, potentially influencing the metabolism and thyroid hormone levels of the Tibetan people. This may be attributed to their metabolic adaptability in their unique high-altitude environment. The potential interference with Vitamin B6 metabolism, particularly related to amino acid metabolism, could have a substantial effect on Tibetan metabolism, given the likelihood of Vitamin B6-rich traditional diets. Steroidal hormones and pantothenate, crucial for life maintenance and immune regulation, also play vital roles in their overall health. Therefore, a scientifically controlled and optimized approach to heat treatment methods for yak milk proteins could enhance their nutritional value and contribute to human health. This study, exploring the relationship between oxidative modification of Yak milk proteins during different heat treatment processes and gut microbiota and metabolism, provides valuable insights from the perspective of protein oxidation.

## 5. Conclusions

The metagenomic analysis of mouse gut microbiota revealed changes in 66 taxonomic genera. In the high-temperature sterilization group (H), prominent differences included Verrucomicrobiales, Verrucomicrobiae, Akkermansiaceae, and 28 others. The high-temperature pasteurization group (M) showed primary changes in genera such as Latilactobacillus, Bacillus, and Romboutsia. The low-temperature pasteurization group (L) exhibited significant variations in genera such as Faecalibacterium, Chaetomium, Paenibacillaceae, Eggerthella, and Sordariales, among 33 others. Functionally, compared to group C, group H upregulated translation and energy metabolism functions, group L enhanced other amino acid metabolism, translation, and cell replication and repair functions, and group M significantly improved other amino acid metabolism, translation, and cell replication and repair functions. Metabolomic analysis indicated differential changes in mouse metabolites affecting multiple metabolic pathways, with the most impacted being phenylalanine metabolism, Vitamin B6 metabolism, steroidal hormone biosynthesis, and the biosynthesis of pantothenate and Coenzyme A. This aligns with the metagenomic analysis of mouse body function pathways, influencing amino acid and energy metabolism. Yak milk holds significant importance in the daily diet of Tibetan people. The findings from this study can assist them in comprehending the effects of various heat treatment methods on yak milk, enabling informed and healthier dietary choices.

## Figures and Tables

**Figure 1 foods-13-00192-f001:**
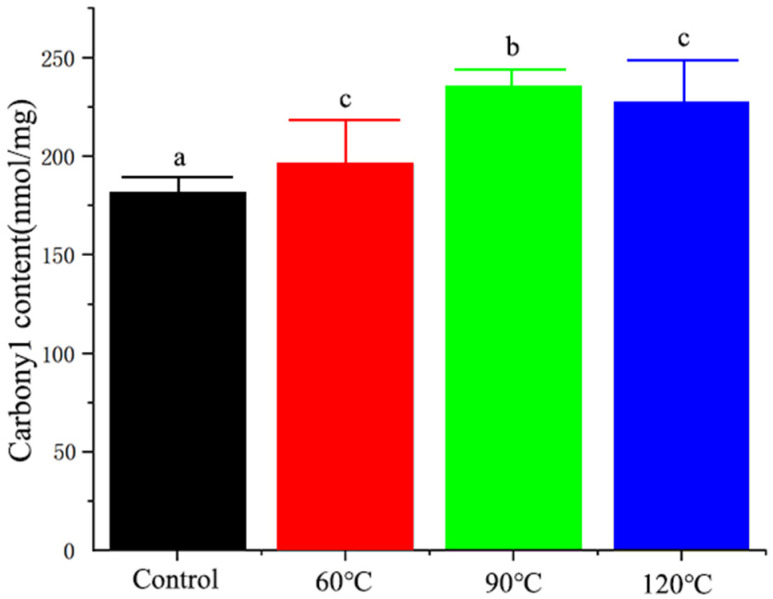
The content of carbonyl groups in yak milk protein treated with different heat treatments. Note: a, b, and c indicate significant differences, and the same degree of difference is indicated by identical letters.

**Figure 2 foods-13-00192-f002:**
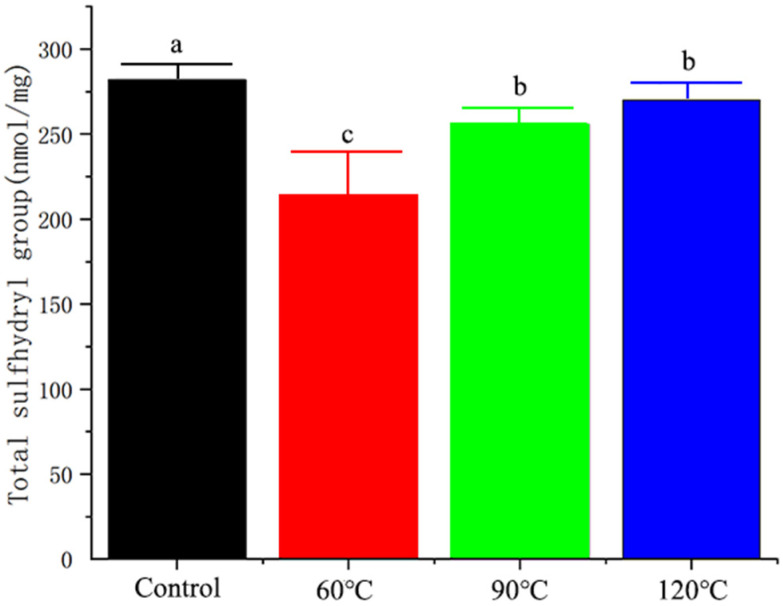
Total sulfol content in yak milk proteins after different heat treatments. Note: a, b, and c indicate significant differences, and the same degree of difference is indicated by identical letters.

**Figure 3 foods-13-00192-f003:**
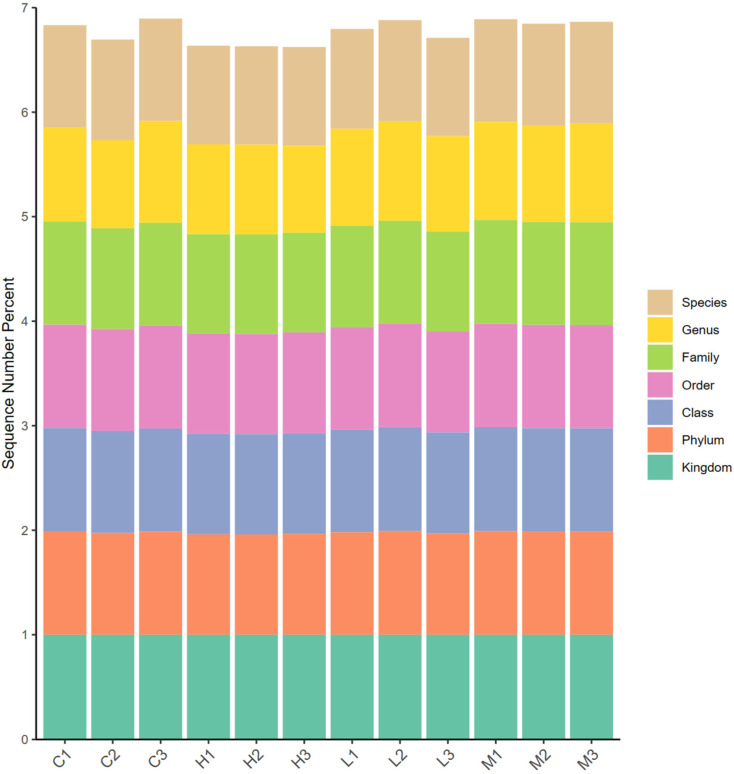
Bar graph of the degree of sequence annotation for each sample at each taxonomic level. Note: The abscissa is the sample name, the ordinate (Sequence Number Percent) represents the ratio of the number of sequences annotated to this level to the total annotation data, and the top-down color order of the bar chart corresponds to the color order of the legend on the right. Each taxonomic level with the highest value of 1, representing 100% of the sequences were annotated at least at this level.

**Figure 4 foods-13-00192-f004:**
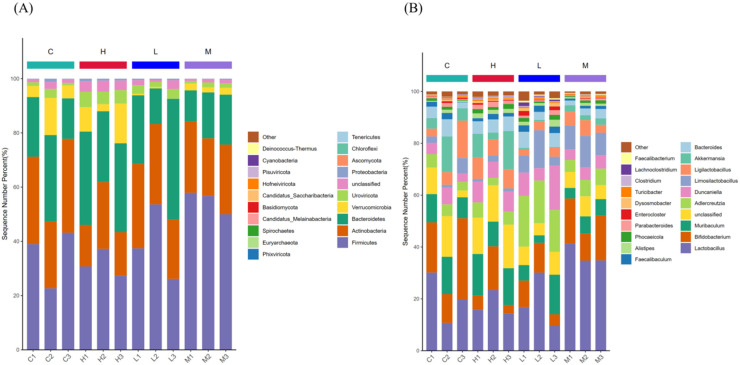
Relative distribution of each sample at the phyla (**A**), genus (**B**) level (top 20 species). Note: The abscissa is the sample name, the ordinate (Sequence Number Percent) represents the ratio of the number of sequences annotated to the phylum/genus level to the total annotation data, and the top-down color order of the bar chart corresponds to the color order of the legend on the right. Sequences without annotation at the phylum/genus level were grouped into the unclassified category. The maximum of 20 species showing the most dominance in the legend, and the remaining species with low relative abundance are classified as Other presented in figure.

**Figure 5 foods-13-00192-f005:**
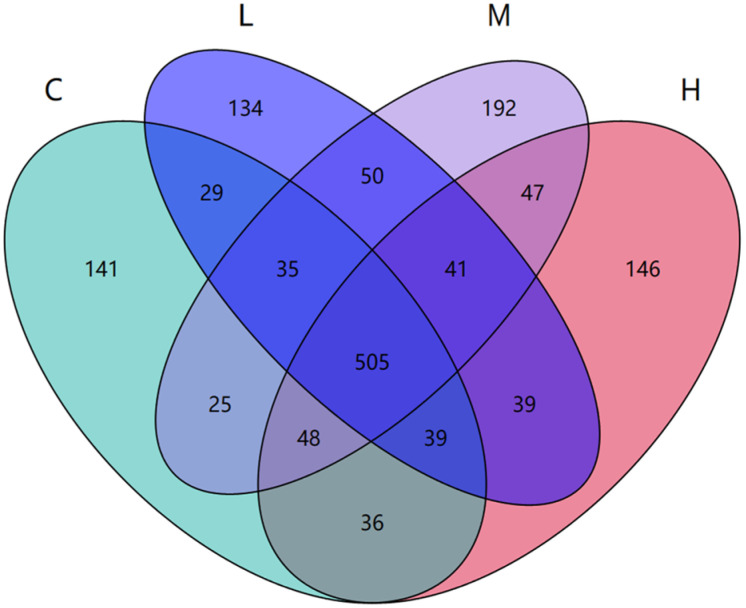
Wayne diagram of common or endemic species. Note: The Wayn diagram shows the number of species shared or unique between different groups, with each ellipse representing a grouping.

**Figure 6 foods-13-00192-f006:**
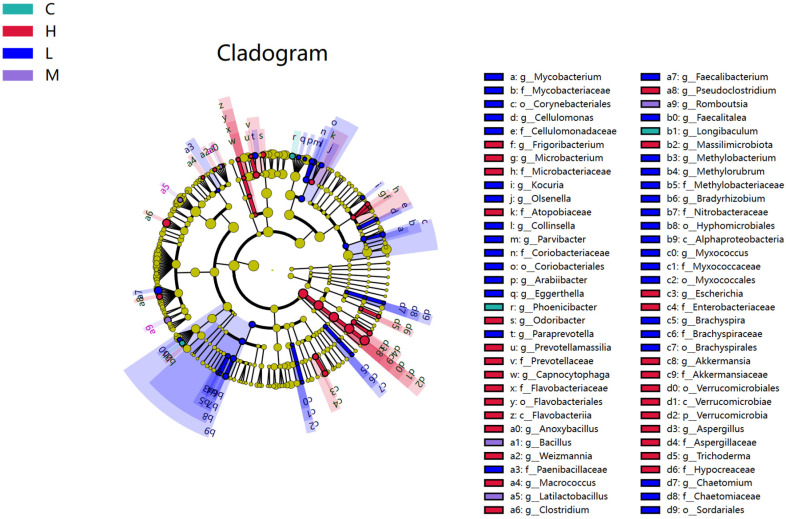
An evolutionary clade plot.

**Figure 7 foods-13-00192-f007:**
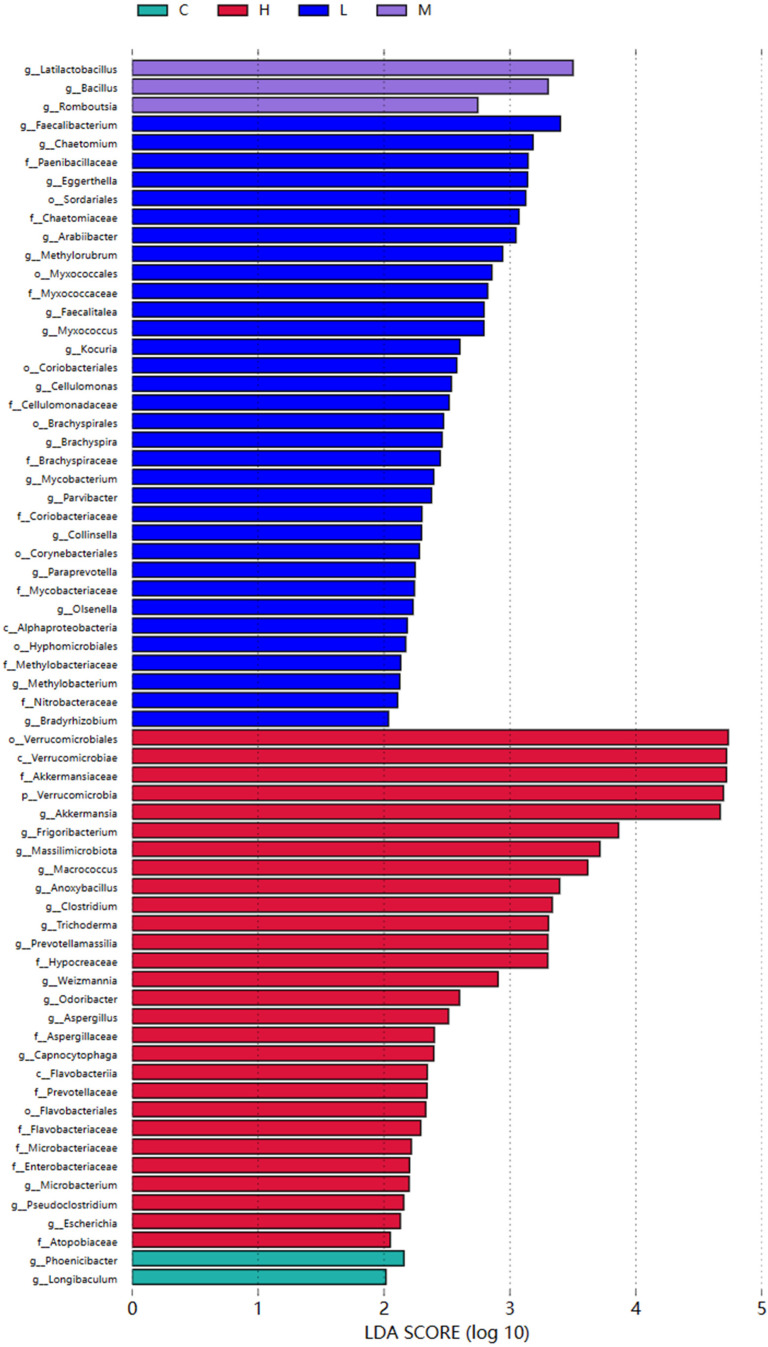
LEfSe analysis of the LDA bar chart. Note: Each lateral column form represents a species, the length of the column form corresponds to the LDA value, the higher the LDA value, the greater the difference. The color pair of the column should be the species of that group, the characteristic microorganism (with relatively high abundance in the corresponding group).

**Figure 8 foods-13-00192-f008:**
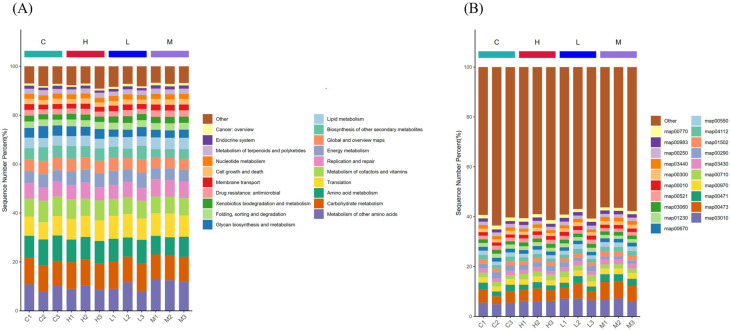
Bar plot of the hierarchical abundance of KEGG metabolic pathways (**A**). LEfSe analysis of the metabolic pathways of KEGG (**B**). Note: In (**A**), the abscissa is the sample name, and the ordinate (Sequence Number Percent) represents the ratio of the relative abundance of the metabolic pathway annotated to each sample at the functional level to the total annotation data. The top-down color order of the bar chart corresponds to the legend color order on the right, the legend shows the most advantageous 20 metabolic pathways, and the remaining metabolic pathways are classified as Other displayed in the figure. In (**B**), the abscissa is the sample name, and the ordinate (Sequence Number Percent) represents the ratio of the relative abundance of functional metabolic pathways to each sample at the functional level the top-down color order of the bar chart corresponds to the color order of the legend on the right, the legend shows the most dominant 20 metabolic pathways, the remaining metabolic pathways are classified as displayed in the figure.

**Figure 9 foods-13-00192-f009:**
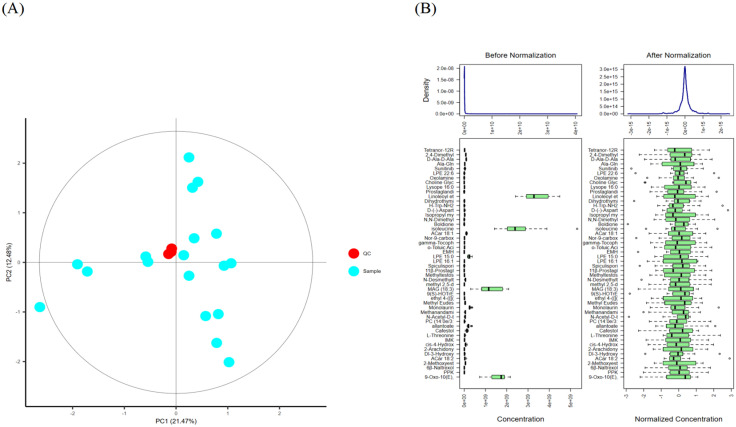
PCA plot of quality control samples (**A**), content distribution of each metabolites before and after standardization correction (**B**). Note: (**A**) The red point is the corrected quality control sample point (QC sample), and the blue point is the test sample. Content distribution is represented by a boxplot, which from left to right correspond to outlier, minimum, lower quartile, median, upper quartile, maximum, outlier. (**B**) shows the distribution before standardization correction, and the right panel shows the distribution after standardization correction.

**Figure 10 foods-13-00192-f010:**
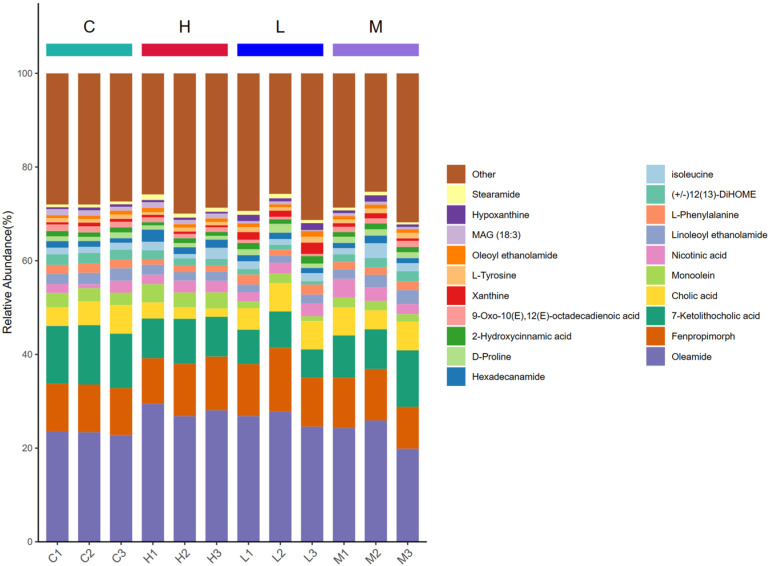
Bar graph of the percentage accumulation of metabolites in the top 20 content. Note: The abscissa is the sample name, sorted according to the grouping order, and different grouping samples are labeled in different colors. The ordinate represents the percent content of each metabolite, and the order of the columns corresponding to the metabolites from the top down is consistent with the figure legend. The figure shows the metabolites ranked in the top 20, and the remaining metabolites are included in Others.

**Figure 11 foods-13-00192-f011:**
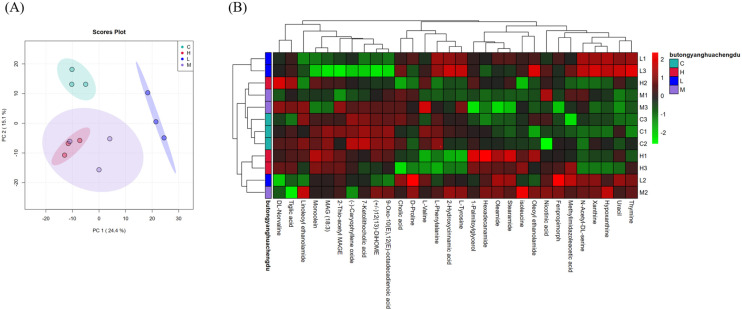
PCA diagram (**A**), and the clustering results of metabolite heat map (**B**). Note: In (**A**) PCA, each point corresponds to one sample, and the distance between two points is approximately the metabolite structure of the two samples (European distance). Different groups are highlighted in different colors, and the area indicated by the ellipse is the 95% confidence region of the sample point. The vertical axis is the sample name information, and it also includes the grouping information. (**B**) Horizontal axis shows the metabolites. The cluster tree at the top of the figure is the similarity cluster of metabolites in each sample, the cluster tree on the left is the sample cluster tree, and the heatmap in the middle is the heat map of metabolite content. The relationship between the color and metabolite content (Z-Score) is shown in the scale at the top right of the figure.

**Figure 12 foods-13-00192-f012:**
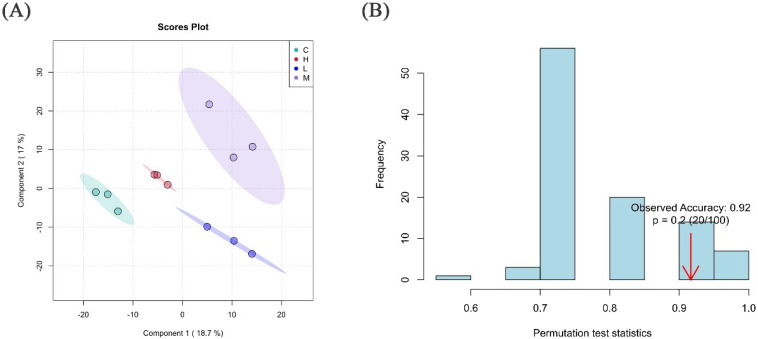
Distribution of test statistics and *p*-value of PLS-DA point cloud map (**A**), PLS-DA displacement test (**B**). Note: In (**A**), each point corresponds to one sample, and the horizontal and vertical coordinates are the values of the two factors with the best discrimination effect. Different groups are highlighted in different colors, and the area indicated by the ellipse is the 95% confidence region of the sample point. (**B**), the abscissa represents the replacement test statistics (model prediction accuracy) between the distribution area, the ordinate is the replacement process of the interval test statistics frequency, the arrow position is the actual observed test statistics value, if the value is far from random distribution, that the model to distinguish the effect is not random, the model to distinguish the effect is significant.

**Figure 13 foods-13-00192-f013:**
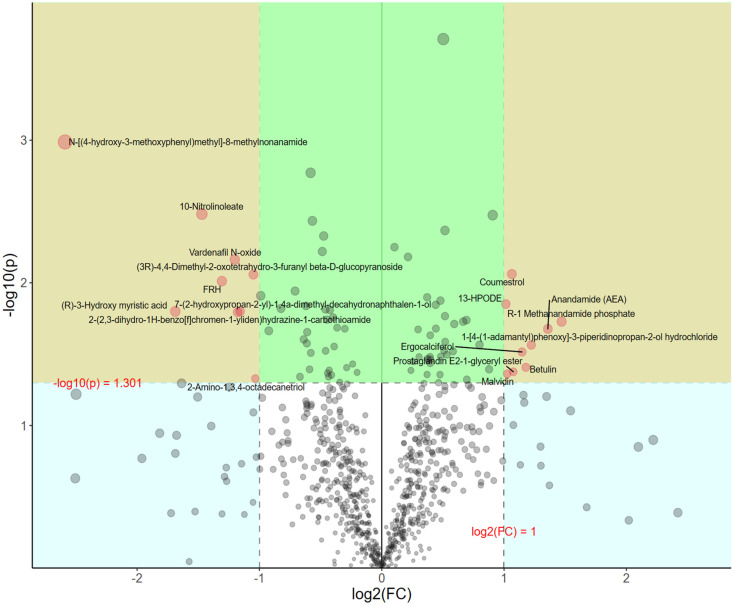
Multiple-change volcano plot. Note: Each point represents a metabolite, the abscissa is the change multiple, and the ordinate is the *p* value of *t* test. The larger the change multiple, the smaller the *p* value (the higher the log10 (p)), the larger the point.

**Figure 14 foods-13-00192-f014:**
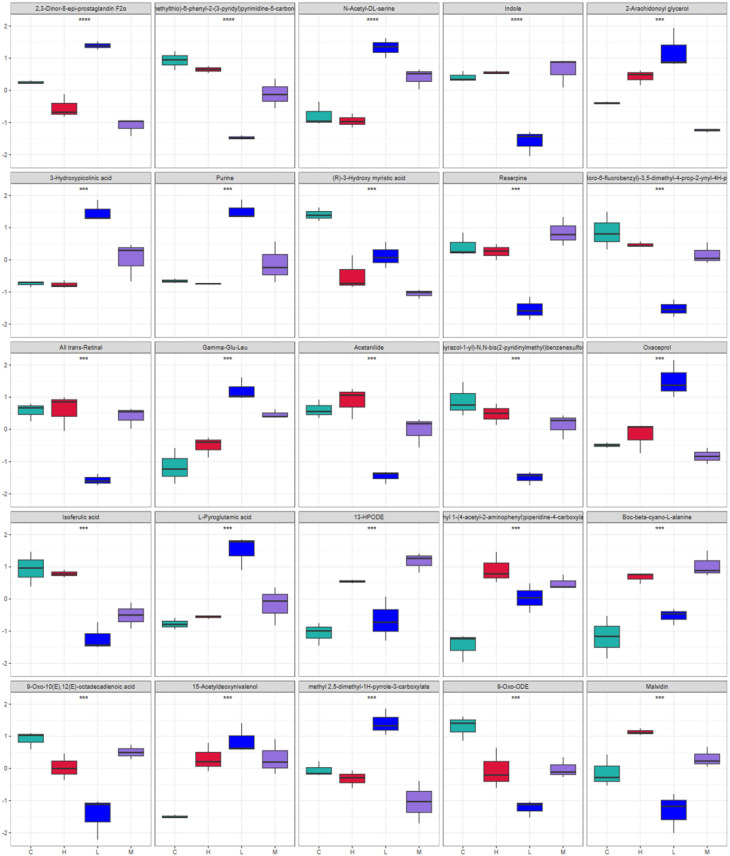
Metabolite difference bin plots (*, ** and *** represent *p* < 0.05, *p* < 0.01 and *p* < 0.001, respectively).

**Figure 15 foods-13-00192-f015:**
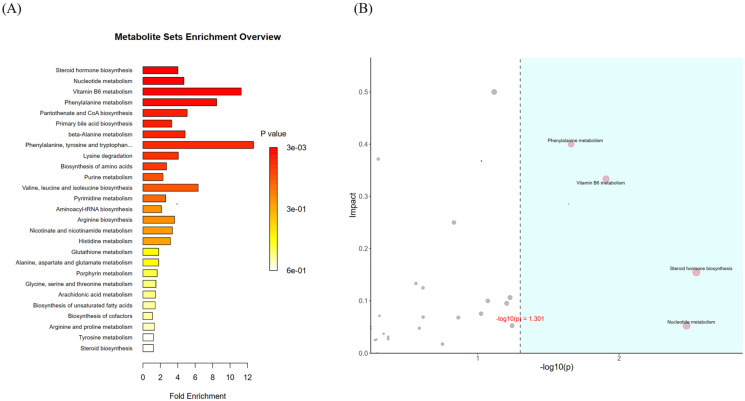
ORA enrichment analysis plot (**A**). ORA enrichment analysis and topology analysis plot (**B**). Note: (**A**) is the enrichment multiple, which is the number of observed metabolites/theoretical metabolites in the metabolic pathway. The magnitude of the *p*-value is indicated by the color, and the darker the color, the smaller the *p*-value. (**B**) The abscissa is the *p*-value of ORA analysis, the blue area is significant (*p* < 0.05); the ordinate is the topology analysis Impact.

**Table 1 foods-13-00192-t001:** Group information of the animal experiments.

Groups	Feeding Patterns	Dosage	Experimental Number
Control group	Normal diet	Free diet	C
Different temperature treatments	65 °C treatment	385.7 mg/kg	L
	90 °C treatment	385.7 mg/kg	M
	120 °C treatment	385.7 mg/kg	H

Note: The optimal adult daily intake of dairy products is 500 mL [25], and the average weight of an adult mouse is 70 kg. Thus, the simulated daily intake for each adult mouse is 385.7 mg/kg.

**Table 2 foods-13-00192-t002:** List of data output quality.

Sample ID	Raw Reads (#)	Raw Base (GB)	%GC	Raw Q20 (%)	Raw Q30 (%)	Clean Reads (#)	Cleaned (%)	Clean Q20 (%)	Clean Q30 (%)
C1	19,676,161	5.9	48	97.12	92.27	18,588,781	94.47	98.36	94.25
C2	23,243,541	6.97	49	97.17	92.36	22,128,761	95.2	98.32	94.15
C3	20,049,616	6.01	49	97.01	92.02	19,011,523	94.82	98.27	94.01
H1	22,853,746	6.86	48	97.13	92.24	21,478,874	93.98	98.31	94.09
H2	19,776,846	5.93	47	97.19	92.43	18,605,722	94.08	98.4	94.36
H3	22,303,799	6.69	48	97.22	92.43	21,327,490	95.62	98.34	94.17
L1	22,337,587	6.7	48	97.16	92.37	21,123,586	94.57	98.38	94.32
L2	20,193,449	6.06	48	97.05	92.14	19,263,463	95.39	98.29	94.09
L3	22,841,107	6.85	49	97.27	92.68	21,779,059	95.35	98.44	94.52
M1	23,299,288	6.99	47	97.08	92.1	22,264,927	95.56	98.25	93.9
M2	23,379,247	7.01	47	97.13	92.23	22,281,520	95.3	98.29	94.05
M3	22,274,283	6.68	48	97.1	92.19	21,013,573	94.34	98.29	94.04

Note: Sample ID: Sample name; Raw Reads (#): Number of sequencing Raw reads; Raw Base (GB): Number of Raw reads in GB, Total number of bases from the sequencing raw data, That is, the number of Raw reads multiplied by the sequencing length; %GC: percentage of G/C bases in the total base number; Clean Reads (#): number of Clean reads obtained after filtering (quality control and host removal sequence); Cleaned (%): percentage of sequences remaining after filtering in Raw reads; Q20, The proportion of bases with a mass score higher than 20; Q30, Proportion of bases with mass scores higher than 30.

## Data Availability

Data are contained within the article.

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
