# Peer review of "Effects of Different Heat Treatments on Yak Milk Proteins on Intestinal Microbiota and Metabolism"

_foods, 2024, doi:10.3390/foods13020192_

Round 1

Reviewer 1 Report

Comments and Suggestions for Authors

The article by Shu et al., investigates the changes in gut microbiota and metabolites due to milk protein oxidation during heat processing, which has potential implications for human gut health. The authors observed notable changes in the gut microbiota composition and metabolic products of mice following the consumption of yak milk proteins subjected to various heat treatments. The manuscript is very well written, nicely organized, and follows standard methodologies.

I have the following minor observations which need to be addressed.  

Abstract:

Lines 34-36: Sentenc can be merged as ....both L and M group significantly upregulated....

Line 43: The last part of the sentence is not clear

Introduction: Please mention how different heat treatment procedures impact the nutritional value of milk protein

Materials and methods:

Line 148: Specify foods in adaptive feeding of animals

Line 154: What normal diet was used? Please specify.

Line 169-170: Please cite the in-text reference as per the journal-specific style

Line 170: Which sample was used? Milk? And what is the diluting factor?

Line 201: For 16s metagenomics, which hypervariable region was targeted?

Line 205: Which sample was used for metabolite extraction?

Section 2.2.5: Details of the sequencing method is needed. 

Results:

Figure 1: Check Tarbonyl in X axis

Check titles of 3.1 and 3.2, maybe chenged to carbonyl and thiol content

Figure 3: Please modify the text in the figure ais and legends such that they can be readable.

Discussion:

Please discuss about any possible role/relationship of differentially abundant bacterial taxa with upregulated/downregulated metabolic pathways including metabolites impacting the pathway in the three groups. This will help in understanding the integrated role of both gut microbiota and metabolites in the metabolism of protein oxidation of heat treatment of milk and overall intestinal health.

Figures: All figures need to be clear. Texts mentioned in the images are not readable.

Author Response

Dear review teacher, the article has modified most of the content according to the requirements, all the images in this article can be viewed according to the uploaded attachment, format content if there are still shortcomings, but also hope the teacher to give valuable advice, thank you very much

Reviewer 2 Report

Comments and Suggestions for Authors

Article: Effects of Different Heat Treatments on Yak Milk Proteins on Intestinal Microbiota and Metabolism

The research is well planned, except for the description of the sample coded M. Ethical rules have been followed. The presentation is beautiful. It is well written but I still have some advice and questions which I have written below and outlined in the article pdf.

1.      The introduction is long.

2.      The heat treatment norm specified with the M code is not the pasteurization norm. The applied period is too long. Therefore, it is more appropriate to refer to it by the applied temperature and time. I wish it had been applied for a short time at this temperature. Because milk produced by the high temperature short time method (HTST method) is practically the most consumed type of drinking milk.

3.      Microorganism names should be written in italic.

4.      In line 75, the question about fatty acids should be answered and corrected.

5.      Table 1, footnote should be answered. In Figure 1, y-axis "Carbonyl" should be written correctly. And Figure 1. Carbonyl content...

6.      Conclusions: As a result, can it be said that the recommended heat treatment for Yak milk is M application? If the applied time had been shorter, would the result have been similar (for example, about 45-60 seconds at 78-80 °C). Would you consider planning to conduct this research under this heat treatment norm?

7.      References: Standard spelling rules were not followed in almost all of the references (Uppercase and lowercase letters, breaks, journal names, unnecessary signs, etc.)

Author Response

(The authors gave the same response as above.)

Reviewer 3 Report

Comments and Suggestions for Authors

The manuscript entitled "Effects of Different Heat Treatments on Yak Milk Proteins on Intestinal Microbiota and Metabolism" is well sctructured. However, some point is needed to attention in the manuscript.

In method section:

- The title of "Test Drugs" is not appropriate. It is material of study. 

- In this part, some verbs are in present forms. It should replace with past tense.

- In section 2.2.1., it is mentioned the centrifugation of yak milk. Which part of milk is used in the study.

- It should mention the equipment of Carbonyl content. How is it measured?

- Title 3.2 and  3.1 are similar

- what are  Q20 and Q30 in table 2 in footnote of table.

Author Response

(The authors gave the same response as above.)

Reviewer 4 Report

Comments and Suggestions for Authors

Please see the comments in the manuscript attached.

Author Response

(The authors gave the same response as above.)
